# An Innovative Personalised Management Program for Older Adults with Parkinson’s Disease: New Concepts and Future Directions

**DOI:** 10.3390/jpm11010043

**Published:** 2021-01-14

**Authors:** Piyush Varma, Lakshanaa Narayan, Jane Alty, Virginia Painter, Chandrasekhara Padmakumar

**Affiliations:** 1Faculty of Medicine, Imperial College London, London SW7 2BU, UK; piyush.varma15@imperial.ac.uk; 2Joint Medical Program, University of Newcastle, Callaghan, NSW 2308, Australia; Lakshanaa.Narayan@uon.edu.au; 3Wicking Dementia Research & Education Centre, School of Medicine, Hobart, TAS 7000, Australia; jane.alty@utas.edu.au; 4School of Medicine, University of Tasmania, Hobart, TAS 7000, Australia; 5Department of Geriatric Medicine, Concord Repatriation General Hospital, Concord, NSW 2139, Australia; virginia.painter@health.nsw.gov.au; 6Centre for Education and Research on Ageing, University of Sydney & Concord Repatriation General Hospital, Concord, NSW 2139, Australia; 7Rankin Park Centre, Department of Geriatric Medicine, Parkinson’s Disease Service for the Older Person, John Hunter Hospital, HNELHD, Newcastle, NSW 2305, Australia

**Keywords:** Parkinson’s disease, non-motor symptoms, carer stress, older persons with Parkinson’s disease, education

## Abstract

Introduction: Parkinson’s disease is a heterogeneous clinical syndrome. Parkinson’s disease in older persons presents with a diverse array of clinical manifestations leading to unique care needs. This raises the need for the healthcare community to proactively address the care needs of older persons with Parkinson’s disease. Though it is tempting to categorise different phenotypes of Parkinson’s disease, a strong evidence based for the same is lacking. There is considerable literature describing the varying clinical manifestations in old age. This article aims to review the literature looking for strategies in personalising the management of an older person with Parkinson’s disease.

## 1. Article Highlight

Education, empowerment, and enablement of a person with Parkinson’s disease (and carers) is the fundamental cornerstone on which personalised medicine for Parkinson’s disease should be based. This article describes a multidisciplinary Parkinson’s disease education program, which received wide acceptance from the Parkinson’s Disease community.

## 2. Introduction

Parkinson’s disease (PD) has historically been considered a motor disorder with cardinal signs of bradykinesia, tremor, rigidity, and postural instability. However, over the last two decades or so, there has been increased recognition of the importance of non-motor symptoms (NMS), including their typical occurrence prior to the onset of the motor presentation [1]. Current understanding now highlights PD as a multisystem heterogeneous disorder [1,2]. Different patterns of motor and NMS lead to the potential for re-categorisation of PD as a syndrome, or syndromes, rather than a singular disease [1,2].

The mainstay of PD management is dopamine replacement therapy (DRT), such as levodopa, which has been shown to improve PD motor symptoms, function, and quality of life [2]. There is evidence that certain types of NMS may be responsive to dopaminergic therapy whilst others can be more refractory. However, there is increasing recognition that the significant variability of NMS and motor symptoms are likely to require a more nuanced approach to address the overall clinical picture rather than the more traditional “one size fits all” approach [2,3].

This is particularly so in older persons with PD where NMS are common, more severe, and cause substantial impairments in quality of life [3]. Furthermore, management of older persons with PD presents a unique challenge when considering comorbidity, medication, and carer burden. Consequently, there has been emerging interest in advancing the personalisation of PD management for older persons, tailored to the specific needs of the individual [2,4]. In this paper, we present a summary of the literature and discuss the advances made in personalised management in the older adult for the treatment of PD. We will describe a novel model of personalised medicine for older adults with PD that has been very popular in the PD patient community in regional New South Wales (NSW), Australia.

## 3. What Is Known about the Subtypes of Parkinson’s Disease?

A vast array of literature highlights that PD presents with a range of different signs and symptoms between individuals. Distinct PD motor subtypes have been described—such as a tremor-dominant subtype (with a slower rate of progression) and a postural instability gait disorder (PIGD) subtype (with a more rapid rate of progression and higher risk of developing dementia).

NMS, such as rapid-eye movement behavioural disorder, cognitive impairment, mood changes, apathy, and fatigue, vary in frequency and severity between persons with PD [1]. Historically, NMS have been poorly recognised and reported leading to inadequate research into their pathophysiology [3]. A clinical tool designed to assess NMS more objectively was only first introduced in 2005 [3].

Some researchers have hypothesised that separate NMS predominant phenotype exists with non-motor PD subtype clusters, such as cardiovascular, mood changes, perception or hallucination, gastrointestinal, urinary, and sexual function [5,6].

However, despite the advances in our understanding of NMS and the usefulness of subtype classification, this may be premature as considerable overlap is seen. There exists contradictory literature regarding the types of NMS experienced by persons within each subtype. Erro et al. [7] demonstrated that patients within the NMS subtypes were more likely to suffer from urinary incontinence, while motor disease subtype resulted more often in neuropsychiatric and cognitive impairment [7]. However, research elsewhere has demonstrated that the NMS subtype predisposes to significant mood and cognitive impairment, while tremor-dominant disease has a relative sparing of NMS [8,9]. One paper noted that cognitive disturbance helped to distinguish NMS from motor disease later in disease, despite nonspecific features of autonomic disturbance featuring more commonly in NMS [7]. These discrepancies may be in part due to the lack of a standardised classification system of PD subtypes and further reflect its heterogeneity.

## 4. How Is Parkinson’s Disease Different in the Older Person?

Clinical phenotype and progression of PD in older persons is different when compared with their younger disease counterparts with increased severity of motor symptoms and a faster rate of decline despite comparable duration of disease [10]. They demonstrate greater severity of NMS such as pain, sleep, cognition, and apathy [10]. In one paper, where persons with PD were characterised into different clusters according to age of onset, the patients in the older subtype had a faster rate of disease progression and were found to experience greater axial instability, bradykinesia, rigidity, and tremor as compared with their younger counterparts, who experienced milder impairment in both motor and cognitive domains [11].

Older persons with PD have increased risk of cognitive impairment, autonomic dysfunction, and visual hallucinations, when compared with their younger disease counterparts [10]. Often, there is greater caregiver burden due to increased NMS burden and problems with adherence with treatment [10,12]. Since PD-associated cognitive impairment increases with chronological age, older persons are at increased risk of PD dementia [9,10]. They are also more likely to develop hallucinations and deterioration in sleep quality secondary to dopaminergic therapies [10]. Therefore, cognitive impairment should be assessed prior to dopaminergic medication prescription for older persons with PD. Deep brain stimulation surgery is generally not recommended in adults over 70 years old due to the risk of surgical complications and potential cognitive side effects [2].

## 5. Personalised Medicine for the Older Person with Parkinson’s Disease

Management of the older PD patient presents unique challenges with distinct differences in clinical phenotype, progression, and treatment considerations. Optimisation of personalised management within the population of older persons with PD must ensure adequate education, enablement, and empowerment of patients and caregivers, respectively [10,12].

Bloem et al. [12] propose that the modern definition of health has moved from referring to complete absence of illness or accompanying social difficulties, to having the skills required to adjust and take control of one’s own condition [12]. Therefore, it is imperative that within a model of patient-centred care, the person with PD (and carers) are given the necessary education regarding their condition and management, so that they can self-manage, and refer to professional advice when required [12].

Educated patients often experience less anxiety associated with their condition [12]. Patients that feel they have greater individual control over their own management, and higher understanding associated with their condition, report increased hope and reduced worry surrounding their illness [13]. This may be especially important in older persons with PD where disease severity, NMS, cognitive impairment, and dopaminergic side effects may adversely affect response to usual pharmacological treatment. Education should extend beyond knowledge of the pathology of the disease, and address the expected lifestyle barriers the patient will experience, as well as how to overcome these [12].

### 5.1. Parkinson’s Disease Multidisciplinary Education Program 

Education, enablement, and empowerment of an older person with PD (and their carer) is the fundamental concept behind the Parkinson’s Disease Programme, which has been operating at the Rankin Park Centre Day Hospital, John Hunter Hospital, in Newcastle, NSW, Australia over the last two decades (Figure 1).

Ten persons with PD and their main carer are invited to attend an 8-week long PD education program offered by the multidisciplinary team based at the Rankin Park Centre Day Hospital in Newcastle, NSW. During weekly sessions, persons with PD and their carers are provided with information and education sessions, group exercise programs, falls prevention strategies, and a comprehensive geriatric assessment is undertaken by a geriatrician. The goal is to provide improved awareness of the diversity of symptoms, better understanding of the NMS burden, and most importantly enabling the person and carer with optimal non-pharmacological strategies to manage. The major issues addressed amongst the patient cohort include falls, anxiety, depression, and dementia-related issues such as hallucinations and carer stress. This intervention has proved widely acceptable and immensely popular amongst the PD community in the local region. Its success firmly emphasizes the need for similar interventions addressing the individualistic needs of older persons with PD.

### 5.2. Comprehensive Geriatric Assessment in the Home

Geriatrician home visit to perform comprehensive geriatric assessment in community-dwelling older persons has been long established in geriatric medicine as a widely accepted and cost-effective means of assessment of the geriatric syndromes. Its benefits on survival and functional outcomes have been demonstrated in the acutely unwell, though evidence is more variable for the community setting. The comprehensive geriatric assessment at home provides a valuable opportunity for first hand assessment of the interaction between the medical and the psychosocial profile of an older person. The carer will feel involved and empowered in delivering the care plans arranged while the service is delivered closer to the person’s home. This model has also been utilised at John Hunter Hospital, in Newcastle, NSW, Australia for older persons with PD, with its underlying philosophy demonstrating great potential for extrapolation to a PD service model in the future.

Consideration of additional comorbidities is essential for both the assessment and management of the older person with PD. Firstly, older persons may present with symptoms that mimic the motor and NMS of PD, but are actually secondary to co-existing conditions. For example, slowed walking due to pain or reduced range of arthritic joint movement, or pain, may mimic a bradykinetic gait [10]. Underlying systemic disease can cause fatigue and contribute to autonomic disturbance [10]. Patients with concurrent prostatic enlargement may also complain of urinary retention and/or incontinence [10]. It is essential to recognise these underlying co-existing conditions as many can be alternatively managed, with subsequent impact on improving quality of life [14]. 

Older persons with PD tend to have a greater number of medications [9]. This polypharmacy poses risks for adverse events, drug-drug interactions, iatrogenic errors, and medication non-adherence, the latter compounded by cognitive impairment. A US study demonstrated that people with PD with poor adherence to treatment were more likely to be hospitalised and require additional care visits at home, compared with those that were adherent [15]. Conversing with the patient to accurately explore any specific barriers to adherence, and using dosette boxes and other aids, may help improve adherence to medication regimens [10]. Patients with a recurrent history of non-adherence may also benefit from pharmacological intervention that is only required once daily, or less [2]. 

Increased incidence of osteoporosis and higher risk of falls means that older persons with PD are at increased risk of injuries and fractures that significantly hinder daily activities of living [12,16]. Therefore, preventative management against osteoporosis and the loss of bone mass must be emphasised, including with advice on calcium intake, effective vitamin D supplementation, and prompt bisphosphonate therapy when indicated [2]. However, to further enhance patient empowerment, measures must be taken to improve the safety of the patient’s daily settings as well. This can be done via use of safety measures within the home, such as handrails and mobility aids, to minimise fall risk [12]. Physiotherapy and careful analysis of the side-effect profile of any coexisting medications are also required to both reduce the risk or extent of osteoporosis, as well as to improve mobility and functionality [12]. Additionally, remote monitoring using sensors and online diaries can be useful in allowing for timely detection of medical problems [12]. Modern technology, such as electronic device typing, also allows clinicians access to information regarding near-falls, so that adaptations can be made to management, before a serious accident occurs [12,17].

### 5.3. Establishing an Individualised Management Plan for Older Persons with PD through Questionnaire

The challenge of designing a tailor-made treatment regime for an older person will start with identifying and prioritising the key NMS for that individual. This is because, unlike for the motor symptoms, there are no well-established and efficacious treatments for the majority of NMS, including dementia.

One method proposed to help empower patients is through use of patient questionnaires [12]. This enables them to report subjective outcomes of treatment measures to their clinician and streamline the shared decision-making process [12]. Patients and clinicians can more clearly see which interventions are beneficial and which may further contribute to patient issues. This is a vital measure for the future, as recent research demonstrated that persons with PD often feel that they are not adequately included in their own management process [12], which can create a barrier to trust in the patient–physician relationship, as well as failing to sufficiently empower the patient. This empowerment must also extend to carers as well, who are immediately involved and whose well-being can have direct impact on the outcomes for older people with PD. Caring for a person with PD can be emotionally and physically difficult, and the burden placed on carers must be addressed, to avoid increased risk of stress, social withdrawal, and mortality associated with the role [12]. These risks increase the likelihood that older persons with PD, who may be unable to independently care for themselves, are admitted to hospital [12]. In 2008, a unified Non-Motor Symptoms Scale (NMSS) addressed the need for simple identification and comprehensive assessment of NMS in persons with PD based on the NMSQ (Non-motor symptoms questionnaire) [18]. The introduction of NMSQ and NMSS were two fundamental steps in the journey towards proactively addressing the needs of an older person with PD. A deeper understanding of the methodology behind these two tools will be the first steps in designing an individualised, person-specific treatment plan for an older person with PD (Figure 2).

## 6. Conclusions

The heterogeneity of PD is becoming more widely acknowledged. In the future, it is likely that PD will be considered a clinical syndrome, or syndromes, rather than a single disease. This has widespread implications, enabling greater patient awareness around their own condition, and ensuring that symptoms normally attributed to PD are more carefully investigated, in order to ensure that misdiagnosis does not occur. The patient experience of PD is highly variable meaning that generalised treatment approaches fail to address individual patient needs and symptoms. Greater strides need to be taken in clinical practice to maximise education, empowerment, and enablement of PD patients, and to streamline the movement to an era of shared decision-making [12]. Older persons with PD require additional nuances to management approach and these need to be addressed in a timely and proactive manner. This paper aimed to outline several measures that healthcare professionals can take in order to make management as personalised to the older PD patient as possible. 

The entire concept of personalised medicine as a template for designing custom made treatment regimens for individual medical conditions is an exciting but at the same time a very early step in medicine.

While it would be the most satisfying step both for the physician and patient alike in the realm of PD, more evidence-based medicine needs to be generated before we can safely design a treatment regime targeting the individual needs of an older person with PD. While it is tempting to generalise the generic principles of managing an older person with a chronic medical condition with emphasis on reducing the polypharmacy and respecting underlying frailty that one expects from an older person with multiple comorbidities, it may not be a rational scientific approach to adopt that approach (Table 1).

The need of the hour is an evidence-based objective approach to answer the question; do we have enough evidence to suggest safe individualised medication regimes to address the medical needs of the older person cohort suffering from Parkinson’s disease?

We do hope that our collective thoughts will be the beginning of an evidence-based answer to that question.

## Figures and Tables

**Figure 1 jpm-11-00043-f001:**
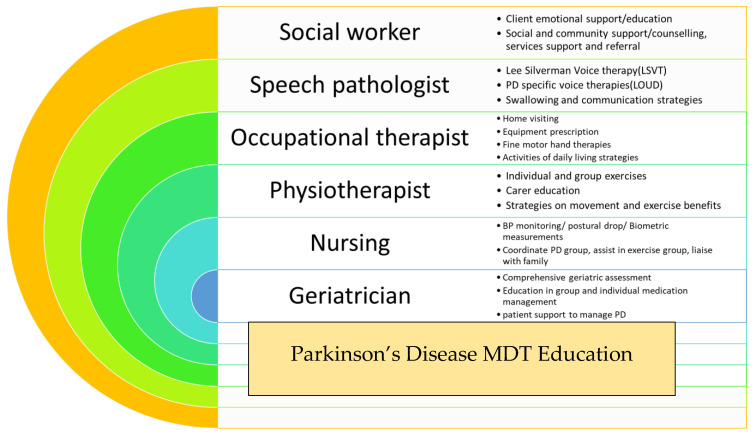
Summary of Parkinson’s disease (PD) programme at Rankin Park Centre Day Hospital, John Hunter Hospital, Newcastle, NSW, Australia.

**Figure 2 jpm-11-00043-f002:**
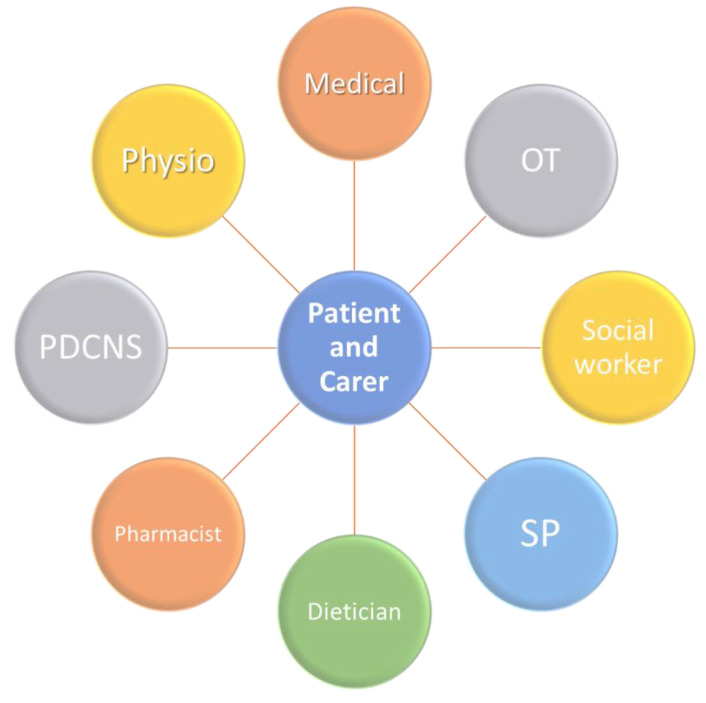
Model of care for the older person with PD. PDCNS: Parkinson’s Disease Clinical Nurse Specialist; SP: Speech Pathologist; OT: Occupational Therapist; Physio: Physiotherapist.

**Table 1 jpm-11-00043-t001:** Take home messages.

Older persons with PD have different care needs compared to a younger person with PD.
We do not have enough evidence-based medicine to identify what those specific care needs are; there is a big scope for future studies in this clinical domain.
There is an increasing amount of evidence advocating the need for monotherapy with L-Dopa in the very frail elderly cohort of people with PD. Whether this is the best approach, we need more studies.
Cognitive impairment plays an important role in deciding which medication should be prescribed in an older person with PD.

## Data Availability

Agree with MDPI research data policies.

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
