# Peer review of "An Innovative Personalised Management Program for Older Adults with Parkinson’s Disease: New Concepts and Future Directions"

_jpm, 2021, doi:10.3390/jpm11010043_

Round 1
Reviewer 1 Report
The article is a comprehensive review of a personalized management protocol for older adults suffering from Parkinson's disease that has been set up in a community in New South Wales, Australia. This review is an accurate description of a collegial strategy to better support older PD patients and their caregivers in their daily management of motor and non motor symptoms. The article does a great job in explaining why we need this type of programs and what are the benefits of this approach in terms of improving the lives of patients and primary caregivers. In particular it seems that a personalized approach may be specifically important in PD since patients may suffer from a large variety of symptoms that together with age and pre-existing conditions can largely affect the patient life expectancy and comorbidity. Therefore as the authors mentioned, I believe it is important to highlight how PD can be described as a spectrum disorder since not all the patients manifest similar symptoms. In line with this it might be worth to mention juvenile forms of PD in the introduction paragraph. In addition in the comparison between old vs young PD patients, what is the average age that the authors associated with old patients? 70y+? It should be specified.
Author Response
The authors would like to thank and acknowledge the comments from Reviewer 1. In response to the comments regarding Juvenile forms of PD , clinical description of Juvenile PD is beyond the remits of this paper. Regarding the average age of older patients , while biological age alone may not be the established marker of Frailty ,Australian Institute of Health and Welfare and Australian Bureau of Statistics ( ABS) have classified people over 65 as Older Australians. Reerence : Australian Law Reform Commission ,Access All Ages- Older Workers and Commonwealth Laws , Report No 120 ( 2013)
Reviewer 2 Report
Please describe in detail the specific method for the innovative individual management program at your facility, and add it if the result was available.
References should be numbered in citation order.
References 15 and 16 do not have citations in the text.
Reference 19 is missing.
Author Response
The authors would like to thanks the Reviewer for noting the errors with the reference numbers and citations.
We would like to inform the Reviewer that the errors have all been reviewed and necessary changes made in the enclosed manuscript.
We do not have a specific method for the innovative individual management program for Parkinson`s Disease in our facility. What we have in place is a Multi Disciplinary Education Programme which has been described in the manuscript along with the Figure. We are hopeful that there is scope for further developing and evolving this Programme into a specific method of innovative individual management programme .